# Syntheses, Reactivities, Characterization, and Crystal Structures of Dipalladium Complexes Containing the 1,3-pyrimidinyl Ligand: Structures of [Pd(PPh_3_)(Br)]_2_(μ,η^2^-C_4_H_3_N_2_)_2_, [Pd(Br)]_2_(μ,η^2^-Hdppa)_2_, and [{Pd(PPh_3_)(CH_3_CN)}_2_(μ,η^2^-C_4_H_3_N_2_)_2_][BF_4_]_2_

**DOI:** 10.3390/molecules25092035

**Published:** 2020-04-27

**Authors:** Hsiao-Fen Wang, Kuang-Hway Yih, Gene-Hsiang Lee

**Affiliations:** 1Department of Hair Styling and Design, Hungkuang University, No. 1018, Sec. 6, Taiwan Boulevard, Shalu Dist., Taichung 43306, Taiwan; 2Department of Applied Cosmetology, Hungkuang University, No. 1018, Sec. 6, Taiwan Boulevard, Shalu Dist., Taichung 43306, Taiwan; khyih0828@gmail.com; 3Instrumentation Center, College of Science, National Taiwan University, Taipei 10617, Taiwan; ghlee@ntu.edu.tw

**Keywords:** dipalladium, pyrimidinyl, tris(pyrazoyl-1-yl)borate, pyrrolidinyldithiocarbamate, doubly bridged, crystal structures

## Abstract

In a refluxing chloroform solution, the η^1^-pyrimidinyl {pyrimidinyl = C_4_H_3_N_2_} palladium complex [Pd(PPh_3_)_2_(η^1^-C_4_H_3_N_2_)(Br)], **1** exhibited intermolecular displacement of two triphenylphosphine ligands to form the doubly bridged η^2^-pyrimidinyl Dipalladium complex [Pd(PPh_3_)(Br)]_2_(μ,η^2^-C_4_H_3_N_2_)_2_, **3**. The treatment of **1** with Hdppa {Hdppa = *N*,*N*-bisdiphenyl phosphinoamine} in refluxing dichloromethane yielded the doubly bridged Hdppa dipalladium complex [Pd(Br)]_2_(μ,η^2^-Hdppa)_2_, **4**. Complex **1** reacted with the bidentate ligand, NH_4_S_2_CNC_4_H_8_ and, NaS_2_COEt, and the tridentate ligand, KTp {Tp = tris(pyrazoyl-1-yl)borate}, to form the η^2^-dithio η^1^-pyrimidinyl complex [Pd(PPh_3_)(η^1^-C_4_H_3_N_2_)(η^2^-SS)], (**5**: SS = S_2_CNC_4_H_8_; **6**: SS = S_2_COEt) and η^2^-Tp η^1^-pyrimidinyl complex [Pd(PPh_3_)(η^1^-C_4_H_3_N_2_)(η^2^-Tp)], **7**, respectively. Treatment of **1** with AgBF_4_ in acetonitrile at room temperature resulted in the formation of the doubly bridged η^2^-pyrimidinyl dipalladium complex [{Pd(PPh_3_)(CH_3_CN)}_2_(μ,η^2^-C_4_H_3_N_2_)_2_][BF_4_]_2_, **8**. All of the complexes were identified using spectroscopic methods, and complexes **3**, **4**, and **8** were determined using single-crystal X-ray diffraction analyses.

## 1. Introduction

C-F activation of a pyrimidine at nickel was reported by Braun [1]. The catalytic activity of dipalladium for Suzuki-Miyaura cross-coupling reactions [1], various organoboronic acids, and aryl bromides by the pyridyl-bridged palladium complex [2] for Suzuki cross-coupling reactions, coupling of aryl boronic acids and aryl halides by the palladacycle [3] complex, the imidazolium palladium complex [4] for Heck reactions, and intramolecular reductive elimination by a Pd-N binuclear complex [5] are crucial in the organic synthesis [6] of palladium complexes catalyzed for forming a C-C bond.

We recently reported syntheses, reactivities, inter- and intramolecular dissociation, and crystal structures of Pd complexes containing thiocarbarmoyl [7], oxythiocarbonyl [8], thiazolinyl [9], and methylpyridinyl [10] moieties. These ligands induced the formation of binuclear complexes that were assisted by the nitrogen or sulfur atom of these ligands. In particular, the sulfur atom of the thiocarbamoyl ligand assists the displacement of either the chloride or the triphenylphosphine ligand to form η^2^-thiocarbamoyl palladium complexes [11] and the thiocarbamoyl-assisted formation of α−form paddlewheel type dipalladium complexes [12].

To obtain a better understanding of how these ligands interact with the metal center, the synthesis of doubly bridged dipalladium complexes, and the application of the pyrimidinyl dipalladium complexes in organic synthesis, we report the syntheses and crystal structures of dipalladium complexes, including the 1, 3-pyrimidinyl containing ligand. Three X-ray crystal structure analyses were conducted to provide structural parameters.

## 2. Results and Discussion

### 2.1. Syntheses

The refluxing chloroform solution of [Pd(PPh_3_)_2_(η^1^-C_4_H_3_N_2_)(Br)], **1** [13] underwent intermolecular displacement of the triphenylphosphine ligand of **1** to form the doubly bridged η^2^-pyrimidinyl dipalladium complex [Pd(PPh_3_)(Br)]_2_(μ,η^2^-C_4_H_3_N_2_)_2_, **3**. However, the intramolecular displacement of the chloride ligand of **1** to form the chelating η^2^-pyrimidinyl complex [Pd(PPh_3_)_2_(η^2^-C_4_H_3_N_2_)][Br], **2** was not observed (Scheme 1). This phenomenon differs from the previously reported complex [Pd(PPh_3_)_2_(η^1^-CSNMe_2_)(Cl)) [7]. In the complex [Pd(PPh_3_)_2_(η^1^-SCNMe_2_)(Cl)], showing inter- and intramolecular dissociation behaviors, the complexes [Pd(PPh_3_)Cl]_2_(μ,η^2^-SCNMe_2_)_2_ and [Pd(PPh_3_)_2_(η^2^-SCNMe_2_)][Cl] [7] were formed. It is plausible that the coordinating ability of the sulfur atom of the thiocarbamoyl ligand assists triphenylphosphine displacement.

Continuously refluxing the chloroform solution of **1** for 3 h produced complex **3** as the final product with 88% isolated yield. No reaction occurred between **1** and Hdppa {*N*,*N*-bis(diphenylphosphino)amine} in dichloromethane at room temperature for 24 h. However, displacement of the pyrimidinyl ligand occurred to form the doubly bridged Hdppa dipalladium complex [Pd(Br)]_2_(μ,η^2^-Hdppa)_2_, **4** with 35% isolated yield in refluxing dichloromethane for 1 h. This also differs from the reaction of [Pd(PPh_3_)_2_{η^1^-CSNMe_2_}Cl] with Hdppa in dichloromethane at room temperature, which was reported to form an α-form paddlewheel dipalladium complex [Pd(μ,η^2^-Hdppa){μ,η^2^-C(S)NMe_2_}]_2_[Cl]_2_ [12].

The treatment of **1** with NH_4_S_2_CNC_4_H_8_ and KTp in dichloromethane at ambient temperature formed an air-stable and yellow η^2^-pyrrolidinyldithiocarbamate complex [Pd(PPh_3_)(η^1^-C_4_H_3_N_2_)(η^2^-S_2_CNC_4_H_8_)], **5** (85%) and η^2^-tris(pyrazoyl-1-yl)borate complex [Pd(PPh_3_)(η^1^-C_4_H_3_N_2_)(η^2^-Tp)], **7** (72%) that released bromide and triphenylphosphine ligands, respectively. Our previous research showed that the reaction of [M(PPh_3_)_2_(η^1^-SCNMe_2_)(Cl)] or [M(PPh_3_)_2_(η^2^-SCNMe_2_)][BF_4_] (M = Pd, Pt) with EtOCS_2_K yielded the Fischer-type carbene-complex [M(PPh_3_){C(SEt)(NMe_2_)}(η^2^-S_2_CO)] (M = Pd, Pt) [14]. The carbene-complex was formed through ethyl migration of the ethyldithiocarbonate ligand to the thiocarbamoyl ligand. To test the generality of this ethyl migration pathway, complex **1** reacted with EtOCS_2_K in dichloromethane at room temperature. Instead of forming the carbene-complex [Pd(PPh_3_){η^1^-C_4_H_3_N(NEt)}(η^2^-S_2_CNC_4_H_8_)], this reaction resulted in the bromide and triphenylphosphine displaced complex [Pd(PPh_3_)(η^1^-C_4_H_3_N_2_)(η^2^-S_2_COEt)], **6** in 70% isolated yield. The complex [Pd(PPh_3_)_2_(η^2^-C_4_H_3_N_2_)][BF_6_], **2** or [Pd(CH_3_CN)(PPh_3_)_2_(η^1^-C_4_H_3_N_2_)][BF_4_], **9** could not be obtained (Scheme 2) from the reaction between **1** and silver tetrafluoroborate, AgBF_4_, in acetonitrile, although the solvent coordination doubly bridged η^2^-pyrimidinyl dipalladium complex [{Pd(PPh_3_)(CH_3_CN)}_2_(μ,η^2^-C_4_H_3_N_2_)_2_][BF_4_]_2_, **8** was formed with 92% isolated yield at ambient temperature. Complex **8** could also be synthesized using the same procedure by reacting AgBF_4_ with complex **3** at room temperature for 2 h.

### 2.2. Infrared Spectroscopy and Mass Spectrometry

In the infrared spectra of **3** and **5**–**8**, the C-N and C-C stretches for the pyrimidinyl group were in the regions of 1537–1567 and 1365–1436 cm^−1^ with partial multiple C-N and C-C bonds, respectively. The peaks at 2288 and 1058 cm^−1^ in the IR spectra of **8** were indicative of C≡N and BF_4_ groups. In the FAB (fast atom bombardment) mass spectra, base peaks with the typical Pd isotope distribution were in accordance with the molecular masses of **3–7**. The FAB mass spectra of **8** showed a parent peak with the typical Pd isotope distribution corresponding to the [M^+^ − 2BF_4_ − 2C_4_H_3_N_2_ − 2CH_3_CN] molecular mass.

### 2.3. Nuclear Magnetic Resonance Spectroscopy

The variable temperature ^31^P{^1^H} NMR spectra of **1** exhibited other sets of triphenylphosphine resonances at δ 29.3 and δ −5.0, which were assigned to the triphenylphosphine of the dipalladium complex [Pd(PPh_3_)(Br)]_2_(μ,η^2^-C_4_H_3_N_2_)_2_, **3** and free PPh_3_, respectively, from 298 to 333 K in CDCl_3_ (Figure 1). Within this temperature range, no doublet resonances were observed in the ^31^P{^1^H} NMR spectra. We can, therefore, conclude that no intramolecular bromide displacement product [Pd(PPh_3_)_2_{η^2^-C_4_H_3_N_2_][Br], **2** was found.

The ^31^P{^1^H} NMR spectra of **3** and **5**–**8** showed singlet resonances in the region of δ 27.8–29.3 which were assigned to the resonance of PPh_3_. The ^31^P{^1^H} NMR spectra of **4** showed a singlet resonance at δ 56.9 for the two Hdppa ligands. In the ^1^H NMR spectra, the 4-H and 5-H protons of the pyrimidinyl group exhibited two singlet resonances in the regions of δ 8.07–8.68 and δ 6.45–6.78, respectively. The ^13^C{^1^H} NMR spectra of **3** and **5**–**8** revealed three singlet resonances in the regions of δ 114–127, δ 154–158, and δ 180–191 which were assigned to the 5-C, 4-C, and 2-C carbon atoms of the pyrimidinyl group, respectively.

The ^1^H NMR spectra of **5**, **6,** and **8** showed eight protons from the S_2_CNC_4_H_8_ ligand exhibiting three multiple resonances at δ 1.93, 3.69, and 3.71, with the integrated ratio being 4:2:2. The δFive protons were present from the S_2_COEt ligand of **6**, exhibiting one triplet resonance at δ 1.41 and one quartet resonance at δ 4.58, with the integrated ratio being 3:2. Six protons were present from the two CH_3_CN ligands of **8**, exhibiting one singlet resonance at δ 1.82. The corresponding ^13^C{^1^H} NMR signals at δ 24.5, 24.6, 49.2, 49.6, at δ 14.1, 53.4, and at δ 2.34, 119.1 were attributed to the S_2_CNC_4_H_8_, S_2_COEt, and CH_3_CN groups, respectively. The ^13^C{^1^H} NMR spectrum of **5** revealed one singlet at δ 205.8 which was assigned to the carbon atom of the dithiocarbamate ligand.

### 2.4. X-ray Single-Crystal Structures of ***3***, ***4***, and ***8***

X-ray single-crystal diffraction studies were conducted on complexes **3**, **4**, and **8**. Single crystals of complexes **3**, **4**, and **8** were grown using a solution containing *n*-hexane/dichloromethane, *n*-hexane/acetone, and acetonitrile solution at 4 °C, respectively. ORTEP (Oak Ridge Thermal-Ellipsoid Plot Program) plots for **3**, **4**, and **8** are shown in Figure 2, Figure 3 and Figure 4, respectively. Table 1 and Table 2 present the crystal data, refinement details, selected bond distances, and angles for **3**, **4**, and **8**. Complexes **3** and **8** were dimers, with each pyrimidinyl unit bridging through a carbon atom of the pyrimidinyl group to one palladium metal center and a nitrogen atom to the other palladium metal center, thus forming a six-membered ring with similar boat-form geometry.

The Pd-Pd bond distance was 3.2008(4) Å in **3** and 3.2275(4) Å in **8,** indicating the absence of bonding between the Pd metal atoms. This is consistent with the previously reported structures [Pd(PPh_3_)Br]_2_{μ,η^2^-C_5_H_3_N(OH)}_2_ (3.1823(9) Å), [Pd(PPh_3_)Br]_2_{μ,η^2^-C_5_H_3_N(NH_2_)}_2_ (3.1925(5) Å) [15], and [Pd(PPh_3_)Br]_2_{μ,η^2^-C_5_H_3_N(CH_3_)}_2_ (3.2254(6) Å), [10]. The pyrimidinyl nitrogen atoms were thus preferentially bonded, and the triphenylphosphine ligands favored the trans site to the nitrogen donors, which have a lower trans influence. In complexes **3** and **8,** the Pd atom and its neighboring atoms lay in a distorted squared plane. A least-squares plane calculation revealed the planarity of the C(1)N(3)Br(2)P(2) and C(5)N(1)Br(1)P(1) cores for **3** (largest deviation 0.1113 and 0.0297 Å, respectively) and the C(5)P(1)N(5)N(1) and C(1)P(2)N(6)N(3) cores for **8** (largest deviation 0.0199 and 0.0631 Å, respectively).

Within the pyrimidinyl ligand itself, the geometry was consistent with a significant partial double bond character in the C-C and C-N bond. The C-N bond distances (1.331(4)~1.358(4) Å) were determined to be typical for a C-N bond with a partial double bond character and much shorter than the normal C-N (1.47 Å) single bond. The average C-C bond distances (1.370(5)~1.378(5) Å for **3** and 1.374(5)~1.378(5) Å for **8**) were comparable to that of the C-C double bond (1.38 Å). The average Pd-C and Pd-N lengths of **3** (1.979, 2.090) and **8** (1.964, 2.089) were in accordance with previously reported values [7,8,9,10,15].

Syntheses of syn and anti-isomers of Pd_2_Cl_2_(dpmMe)_2_ [16], as well as the phosphine and arsine Pd(I) dimers Pd_2_(dppm)_2 × 2_ (X = Cl, Br) and Pd_2_(dam)_2_Cl_2_ {dam = bis(diphenylarsino)methane}, were originally provided by Colton and coworkers [16]. Complex **4** has not been previously prepared.

As shown in Figure 3, a least-squares plane calculation revealed the planarity of the P(2)P(4)Br(2)Pd(1) and P(1)P(3)Br(1)Pd(2) cores for **4** (largest deviation 0.0703 and 0.0534 Å). The coordination geometry around each Pd atom was approximately square planar, with all angles about the Pd center being within ± 7° of 90°. Each approximate square plane included two transoid Hdppa phosphines: A bromine atom, and a Pd-Pd interaction. Complex **4** is a compound containing palladium(I). The trans angles exhibited by these atoms were P-Pd-P (av) = 173.93 (5)^o^ and Br-Pd(1)-Pd(2) (av) = 176.54 (2)^o^. As expected, the phosphorus atoms exhibited an almost tetrahedral geometry. The largest deviation from idealized tetrahedral phosphorus was C(13)-P(2)-Pd(2) = 119.24 (17)^o^. The mean P-N-P angle through the Hdppa amine nitrogen was 115.1 (3)^o^. The Pd-Pd separation of 2.6414 (6) Å is well within the range normally reported for Pd-Pd single bonds. It falls between those values observed for the unbridged complex Pd_2_(CNMe)_6_^2+^ (2.5310 (9) Å) [17] and bridged complexes Pd_2_Br_2_(dppm)_2_ (2.699 Å) [17].

The mean planes through Pd1-Pd2-P1-P2-N1 and Pd1-Pd2-P3-P4-N2 forms a dihedral angle of 32.662°. It was significantly smaller than that found in similar complexes, such as Pd_2_Br_2_(dppm)_2_ (39°) and Pd_2_Br_2_(dmpm)_2_ (50.5°) [17]. The remaining bond distances were normal. The mean Pd-Br distance of 2.5170(7) Å in **4** was slightly shorter than the mean Pd-Br distance in Pd_2_Br_2_(dppm)_2_ (2.535 (6) Å). The mean Pd-P distance was 2.2809(14) Å, which compared favorably with that of Pd_2_Br_2_(dppm)_2_ (2.288 (1) Å) and Pd_2_Br_2_(dmpm)_2_ (2.283 (3) Å). There were no unusual intermolecular contacts.

## 3. Materials and Method

### 3.1. Materials

All manipulations were performed under nitrogen using vacuum-line, drybox, and standard Schlenk techniques. NMR spectra were recorded on a Bruker AM-500 WB FT-NMR spectrometer and are reported in units of δ (ppm) with residual protons in the solvent as an internal standard (CDCl_3_, δ 7.24). IR spectra were measured on a Nicolate Avator-320 instrument and were referenced to a polystyrene standard, using cells equipped with calcium fluoride windows. Mass spectra were recorded on a JEOL SX-102A spectrometer. Solvents were dried and deoxygenated by refluxing over the appropriate reagents before use. The *n*-Hexane, diethyl ether, THF (tetrahydrofuran), and benzene were distilled from sodium-benzophenone. Acetonitrile and dichloromethane were distilled from calcium hydride. Methanol was distilled from magnesium. All other solvents and reagents were of reagent grade and were used as received. Elemental analyses and X-ray diffraction studies were carried out at the Regional Center of Analytical Instrumentation located at the National Taiwan University. PdCl_2_·xH_2_O was purchased from Strem Chemical, C_4_H_8_NCS_2_NH_4_, EtOCS_2_Na, Hdppa, and AgBF_4_ were purchased from Merck.

#### 3.1.1. [Pd(PPh_3_)(Br)]_2_(μ,η^2^-C_4_H_3_N_2_)_2_, **3**

CHCl_3_ (20 mL) was added to a flask (100 mL) containing [Pd(PPh_3_)_2_(η^1^-C_4_H_3_N_2_)Br], **1** (0.789 g, 1.0 mmol). The solution was refluxed for 3 h then diethyl ether (30 mL) was added to the solution and a pale-yellow precipitate was formed. The precipitate was collected by filtration (G4) washed with *n*-hexane (2 × 10 mL) and then dried in vacuo yielding 0.464 g (88%) of **3**. Spectroscopy for **3**: IR (KBr, cm^−1^) υ(CN) 1539(m), 1563(m). ^31^P{^1^H} NMR (202MHz, CDCl_3_, 298K): δ 29.3 (s, PPh_3_). The ^1^H NMR (500MHz, CDCl_3_, 298K): δ 6.46, 6.57 (br, 2H, 5-H of pyrimidine), 7.27–7.86 (m, 30H, PPh_3_), 8.62, 8.68 (br, 4H, 4-H of pyrimidine). ^13^C{^1^H} NMR: δ 116.2, 116.6 (s, 5-C of pyrimidine), 128.2–128.5 (m, o-C of Ph), 130.2, 130.6, 130.8 (m, p-C of Ph), 134.7, 134.8, 134.9 (m, m-C of Ph), 131.9–132.1 (s, 4-C of pyrimidine). MS (FAB, CHCA, *m/z*): 817 [M^+^ − 2C_4_H_3_N_2_ − Br]. Anal. Calcd for C_44_H_36_Br_2_N_4_P_2_Pd_2_: C, 50.08; H, 3.44; N, 5.31%. Found: C, 50.28; H, 3.58; N, 5.20.

#### 3.1.2. [Pd(Br)]_2_(μ,η^2^-Hdppa)_2_, **4**

CH_2_Cl_2_ (40 mL) was added to a flask (100 mL) containing **1** (0.789 g, 1.0 mmol) and Hdppa (0.384 g, 1.0 mmol) and the solution was stirred at refluxing temperature. After refluxing 1 h, MeOH (10 mL) was added to the solution and the yellow solids were formed which were isolated by filtration (G4), washed with *n*-hexane (2 × 10 mL), and subsequently dried under vacuum yielding 0.42 g (35%) **4**. IR (KBr, cm^−1^) υ(NH) 3181(m). ^1^H NMR (500MHz, CDCl_3_ 298K): δ7.14–7.76 (m, 30H, Ph). ^31^P{^1^H} NMR (202MHz, CDCl_3_, 298K): δ 56.9 (s, dppa). ^13^C{^1^H} NMR (125MHz, CDCl_3_, 298K): δ 128.2–137.2 (m, C of Ph) MS (FAB, CHCA, *m/z*): 1141 [M^+^], 1061 [M^+^ − Br], 982 [M^+^ − 2Br]. Anal. Calcd for C_48_H_42_Br_2_N_2_P_4_Pd: C, 50.42; H, 3.70; N, 2.45%. Found: C, 50.04; H, 3.46; N, 2.28.

#### 3.1.3. [Pd(PPh_3_)(η^1^-C_4_H_3_N_2_)(η^2^-S_2_CNC_4_H_8_)], **5**

CH_2_Cl_2_ (20 mL) was added to a flask (100 mL) containing **1** (0.789 g, 1.0 mmol) and NH_4_S_2_CNC_4_H_8_ (0.164 g, 1.0 mmol) at ambient temperature. The mixture was stirred for about 3 days. The mixture was filtered by filtration (G4) with celite and the solution was concentrated to 10 mL, and 20 mL of diethyl ether was added to the solution. The light-yellow solids were formed, which were isolated by filtration (G4), washed with *n*-hexane (2 × 10 mL) and subsequently dried under vacuum yielding 0.493 g (85%) of [Pd(PPh_3_)(η^1^-C_4_H_3_N_2_)(η^2^-S_2_CNC_4_H_8_)], **5**. Spectroscopic data for **5**: IR (KBr, cm^−1^) υ(CN) 1538(m), 1550(m). ^31^P{^1^H} NMR: δ 27.8 (s, PPh_3_). ^1^H NMR: δ 1.93 (m, 4H, NCH_2_C*H_2_*), 3.69, 3.71 (m, 4H, NC*H_2_*CH_2_), 6.45 (s, 1H, 5-H of pyrimidine), 7.27–7.70 (m, 15H, PPh_3_), 8.07 (s, 2H, 4-H of pyrimidine). ^13^C{^1^H} NMR: δ 24.5, 24.6 (s, NCH_2_*C*H_2_), 49.2, 49.6 (s, N*C*H_2_CH_2_), 114.8 (s, 5-C of pyrimidine), 128.6 (m, o-C of Ph), 130.1 (m, p-C of Ph), 134.2, 134.6 (m, m-C of Ph), 153.8, 158.0 (s, 4-C of pyrimidine), 191.5 (s, 2-C of pyrimidine), 205.8 (s, S_2_CN). MS (FAB, NBA, *m/z*): 446 [M^+^ – S_2_CNC_4_H_8_]. Anal. Calcd. for C_27_H_26_N_3_PS_2_Pd: C, 54.59; H, 4.41; N, 7.08. Found: C, 54.65; H, 4.38; N, 7.18.

#### 3.1.4. [Pd(PPh_3_)(η^1^-C_4_H_3_N_2_)(η^2^-S_2_COEt)], **6**

CH_2_Cl_2_ (20 mL) was added to a flask (100 mL) containing **1** (0.789 g, 1.0 mmol) and NaS_2_COEt (0.167 g, 1.0 mmol) at ambient temperature. The mixture was stirred for about 3 h. The mixture was filtered by filtration (G4) with celite and the solution was concentrated to 10 mL, and 20 mL of diethyl ether was added to the solution. The light-yellow solids were formed, which were isolated by filtration (G4), washed with *n*-hexane (2 × 10 mL), and subsequently dried under vacuum yielding 0.398 g (70%) of [Pd(PPh_3_)(η^1^-C_4_H_3_N_2_)(η^2^-S_2_COEt)], **6**. Spectroscopic data for **6**: IR (KBr, cm^−1^) υ(CN) 1534(m), 1546(m). ^31^P{^1^H} NMR: δ 29.5 (s, PPh_3_). ^1^H NMR: δ 1.41 (t*,* 3H, OCH_2_C*H_3_*, ^3^*J*_H-H_ = 8.0), 4.58 (q, 2H, OC*H_2_*CH_3_), 6.44 (s, 1H, 5-H of pyrimidine), 7.37–7.62 (m, 15H, Ph), 8.15 (d, 2H, 4-H of pyrimidine, ^5^*J*_P-H_ = 3.56). ^13^C{^1^H} NMR: δ 14.1 (s, OCH_2_*C*H_3_), 53.4 (s, O*C*H_2_CH_3_), 114.4 (s, 4-C of pyrimidine), 128.3 (m, o-C of Ph), 130.4 (m, p-C of Ph), 133.9 (m, m-C of Ph), 155.5 (s, 5-C of pyrimidine), 182.4 (s, 2-C of pyrimidine). MS (FAB, NBA, *m/z*): 479 [M^+^ − SOEt]. Anal. Calcd. for C_25_H_23_N_2_OPS_2_Pd: C, 52.77; H, 4.08; N, 4.93. Found: C, 52.65; H, 4.18; N, 4.78.

#### 3.1.5. [Pd(PPh_3_)(η^1^-C_4_H_3_N_2_)(η^2^-Tp)], **7**

CH_2_Cl_2_ (20 mL) was added to a flask (100 mL) containing **1** (0.789 g, 1.0 mmol) and KTp (0.277 g, 1.1 mmol). The solution was stirred for 3 h. The mixture was filtered by filtration (G4) with celite then *n*-hexane (30 mL) was added to the solution and a light-yellow precipitate was formed. The precipitate was collected by filtration (G4) washed with *n*-hexane (2 × 10 mL) and then dried in vacuo yielding 0.474 g (72%) of **7**. Spectroscopic data for **7**: IR (KBr, cm^−1^) υ(CN) 1537(m), 1547(m). ^31^P{^1^H} NMR: δ 27.8 (s, PPh_3_). ^1^H NMR: δ 6.05 (s, 2H, 3,5-H of pyrazole), 6.44 (t, 2H, 4-H of pyrimidine, ^3^*J*_H-H_ = 4.8), 6.96, (s, 1H, 4-H of pyrazole), 7.27–7.36 (m, 15H, PPh_3_), 7.44, 7.62 (m, 2H, 4-H of pyrazole), 7.64 (m, 2H, 5-H of pyrazole), 7.86 (m, 2H, 3-H of pyrazole), 8.06 (d, 1H, 5-H of pyrimidine, ^3^*J*_H-H_ = 4.8). ^13^C{^1^H} NMR: δ 115.5 (s, 5-C of pyrimidine), 128.3, 128.4 (m, o-C of Ph), 130.4 (s, 4-C of pyrazole), 131.9–132.1 (m, p-C of Ph), 134.2 (m, m-C of Ph), 136.1 (s, 5-C of pyrazole), 141.3 (s, 3-C of pyrazole), 154.6 (s, 4-C of pyrimidine), 185.8 (d, 2-C of pyrimidine, ^2^*J*_P-C_ = 3.4). MS (FAB, NBA, *m/z*): 660 [M^+^], 593 [M^+^ − pyrazole]. Anal. Calcd. for C_31_H_27_BN_8_PPd: C, 56.43; H, 4.13; N, 16.99. Found: C, 56.68; H, 4.01; N, 17.15.

#### 3.1.6. [{Pd(PPh_3_)(CH_3_CN)}_2_(μ,η^2^-C_4_H_3_N_2_)_2_][BF_4_]_2_, **8**

Acetonitrile (20 mL) was added to a flask (100 mL) containing AgBF_4_ (0.195 g, 1.0 mmol) and **1** (0.789 g, 1.0 mmol). The solution was stirred for 2 h at room temperature. The solvent was then removed under vacuum. The remaining solid was dissolved in 10 mL of CH_2_Cl_2_ and the solution was filtered to remove excess AgBF_4_. The solution was concentrated under vacuum and *n*-hexane (10 mL) was added to initiate precipitation. The pale-yellow solids **8** were formed, which were isolated by filtration (G4), washed with *n*-hexane (2 × 10 mL), and subsequently dried under vacuum yielding **8** (0.529 g, 92%). Further purification was accomplished by recrystallization from diethyl ether/acetonitrile. Spectroscopic data of **8** were as follows. IR (KBr, cm^−1^) υ(CN) 1541(m), 1567(m); υ(BF_4_) 1058(vs). ^1^H NMR (500MHz, CDCl_3_, 298K): δ 1.82 (s, 6H, CH_3_CN), 6.78 (t, 2H, 5-H of pyrimidine, ^6^*J*_P-H_ = 4.04), 7.27–7.70 (m, 30H, PPh_3_), 7.67, 8.52 (t, 4H, 4-H of pyrimidine, ^5^*J*_P-H_ = 1.72, 2.16). ^31^P{^1^H} NMR (202MHz, CDCl_3_, 298K): δ 28.7 (s, PPh_3_). ^13^C{^1^H} NMR (125MHz, CDCl_3_, 298K): δ 2.34 (s, CH_3_), 119.1 (s, CN), 127.4, 127.8 (s, 5-C of pyrimidine), 129.2, 129.3 (s, o-C of Ph), 130.4 (s, 4-C of pyrazole), 131.9 (s, p-C of Ph), 133.8, 133.9 (s, m-C of Ph), 156.3, 158.9 (s, 4-C of pyrimidine), 180.5 (t, 2-C of pyrimidine, ^2^*J*_P-C_ = 4.10). MS (FAB, CHCA, *m/z*): 944 [M^+^ − BF_4_− C_4_H_3_N_2_ − CH_3_CN], 738 [M^+^ − 2BF_4_– 2C_4_H_3_N_2_ – 2CH_3_CN]. Anal. Calcd. for C_48_H_42_B_2_F_8_N_6_P_2_Pd_2_: C, 50.08; H, 3.68; N, 7.30%. Found: C, 50.28; H, 3.86; N, 7.51.

Complex **8** can also be synthesized using the same procedure by employing AgBF_4_ with complex **3**.

### 3.2. X-ray Crystallography

#### Single-Crystal X-ray Diffraction Analyses of 3, 4, and 8

Single crystals of **3**, **4**, and **8** suitable for X-ray diffraction analyses were grown by recrystallization from 20/1 *n*-hexane/dichloromethane, *n*-hexane/acetone, and diethyl ether/acetonitrile, respectively. The diffraction data were collected at room temperature on an Enraf-Nonius CAD4 diffractometer equipped with graphite-monochromated Mo Kα (λ = 0.71073 Å) radiation. The raw intensity data were converted to structure factor amplitudes and their esds after corrections for scan speed, background, Lorentz, and polarization effects. An empirical absorption correction, based on the azimuthal scan data, was applied to the data. Crystallographic computations were carried out on a Microvax III computer using the NRCC-SDP-VAX structure determination package [18].

A suitable single crystal of **3** was mounted on the top of a glass fiber with glue. Initial lattice parameters were determined from 24 accurately centered reflections with θ values in the range from 1.89 to 27.50°. Cell constants and other pertinent data were collected and are recorded in Table 1. Reflection data were collected using the θ/2θ scan method. Three check reflections were measured every 30 min throughout the data collection and showed no apparent decay. The merging of equivalent and duplicate reflections gave a total of 28,150 unique measured data, of which 9862 reflections with *I* > 2σ(*I*) were considered observed. The first step of the structure solution used the heavy-atom method (Patterson synthesis), which revealed the positions of metal atoms. The remaining atoms were found in a series of alternating difference Fourier maps and least-squares refinements. The quantity minimized by the least-squares program was *w*(|Fo|–|Fc|)^2^, where *w* is the weight of a given operation. The analytical forms of the scattering factor tables for the neutral atoms were used [19]. The non-hydrogen atoms were refined anisotropically. Hydrogen atoms were included in the structure factor calculations in their expected positions on the basis of idealized bonding geometry but were not refined in least squares. All hydrogens were assigned isotropic thermal parameters 1.2 Å^2^ larger than the equivalent ***B***iso value of the atom to which they were bonded. The final residuals of this refinement were ***R*** = 0.041 and ***Rw*** = 0.084.

The procedures for **4** and **8** were similar to those for **3**. Selected bond distances and angles are listed in Table 2. X-ray crystallographic files, in CIF (crystallographic information file) format, for the structures of complexes **3**, **4**, and **8** are given in the Appendix A.

## 4. Conclusions

In this article, we report the syntheses, dissociation behavior, and X-ray crystal structures of pyrimidinyl Pd complexes. The pyrimidinyl **1** is a useful material with which to begin synthesizing doubly bridged pyrimidinyl dipalladium complexes. In complexes **3** and **8**, each pyrimidinyl unit bridged through the carbon atom of the pyrimidinyl group to one palladium metal center and nitrogen atom to the other palladium metal center, forming a six-membered ring with similar boat-form geometry. In complexes **5**–**7**, the pyrimidinyl ligand coordinated through the carbon atom to the metal center and the dithiocarbamate, ethyldithiocarbonate, and tris(pyrazoyl-1-yl)borate ligand coordinated to the Pd atom by chelating through two sulfur and two nitrogen atoms, respectively. Regarding the formation time of the dipalladium complexes, including the doubly bridged ligands, the sulfur-containing ligands (thiocarbarmoyl, oxythiocarbonyl, thiazolinyl) were faster than the nitrogen-containing ligands (3-hydroxypyridinyl, 3-aminopyridinyl, methylpyridinyl, and 1,3-pyrimidinyl).

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
