# Peer review of "Syntheses, Reactivities, Characterization, and Crystal Structures of Dipalladium Complexes Containing the 1,3-pyrimidinyl Ligand: Structures of [Pd(PPh3)(Br)]2(μ,η2-C4H3N2)2, [Pd(Br)]2(μ,η2-Hdppa)2, and [{Pd(PPh3)(CH3CN)}2(μ,η2-C4H3N2)2][BF4]2"

_molecules, 2020, doi:10.3390/molecules25092035_

Round 1

Reviewer 1 Report

The manuscript „Syntheses, Reactivities, Characterization, and Crystal Structures of Dipalladium Complexes Containing the 1,3-pyrimidinyl Ligand ...“ by Hsiao-Fen Wang, Kuang-Hway Yih and Gene-Hsiang Lee is solid scientific work. The authors report metalorganic synthesis that allows access to interesting bridged dinuclear and mononuclear palladium compounds. Therefore the manuscript will have its readers and is worth publishing. I rate the manuscript “minor revision” as the title should be carefully rethought if not a more readable and perhaps a bit more catchy one could be used instead. Second the Abstract should be rewritten. Why? Molecules is an open source journal. Everyone can download the pdf-file and read the whole manuscript therefore there is no necessity to explain everything like “reflux conditions in chloroform” in the abstract. A more inviting abstract, that makes more curious for the whole manuscript would be preferable. Additionally should Hdppa be explained in the abstract, as it used there.

I miss an explanation of KTp ind the text.

In 2.4 the authors stated, “To obtain a definitive result…” sorry that statement is in my opinion wrong, as the statement is only valid for the situation inside the crystal(s) they investigated. Please use a formulation that is more sensitive to that often-made mistake, that a x-ray structure is equivalent with the absolute truth.

Note (Personal statement that was not considered for rating the manuscript):

I often get badly written manuscript with poor chemistry for reviewing, but they are often written in away that the reader is fascinated for the described chemistry and feels uncomfortable to rate the manuscript “reject”. Here the situation is different, if I wouldn’t have been a reviewer I would have stopped reading, as I am currently not planning to do the described synthesis. I personally miss the enthusiasm of the authors for their chemistry. Please think about if you can improve the manuscript in that direction, too. As this is not a scientific argument, as it is only an emotional I have not considered that point in my review.

Author Response

Q1:The title should be carefully rethought if not a more readable and perhaps a bit more catchy one could be used instead.

A1:We would like to maintain the original title because it is clearly and fit the contents of the MS. We hope it is acceptable. We also accept the revised title by the editor-in-chief.

Q2:Second the Abstract should be rewritten. Why? Molecules is an open source journal. Everyone can download the pdf-file and read the whole manuscript therefore there is no necessity to explain everything like “reflux conditions in chloroform” in the abstract. A more inviting abstract, that makes more curious for the whole manuscript would be preferable.

A2:Line 15-27: The abstract has been revised for clearly. We would like to keep the original sentence like “In a refluxing chloroform solution” because this is a very important condition for the intermolecular displacement reaction and it is interesting to the readers. Hope this thinking is acceptable.

Q3:Additionally should Hdppa be explained in the abstract, as it used there.

A3: Line 18-19: The Hdppa {Hdppa = N, N-bisdiphenyl phosphinoamine} has been explained in the abstract of the revised MS.

Q4:I miss an explanation of KTp ind the text.

A4:Line 21: The tridentate ligand, KTp {Tp = tris(pyrazoyl-1-yl)borate}, has been explained in the abstract.

Q5:In 2.4 the authors stated, “To obtain a definitive result…” sorry that statement is in my opinion wrong, as the statement is only valid for the situation inside the crystal(s) they investigated. Please use a formulation that is more sensitive to that often-made mistake, that a x-ray structure is equivalent with the absolute truth.

A5:Line 126: The “To obtain a definitive result” has been deleted in the revised MS.

We thank three reviewers’s comments for the improvement of the MS and We hope the revised MS is acceptable for publication in your journal.

Best Regards,

Dr. H.F. Wang

Reviewer 2 Report

This paper explores the reactivity of a square pyramidal palladium(II) complex with two triphenylphospine, bromide and η1-pyrimidine (complex 1) with a variety ligands, in particular potentially bidentate ligands, including pyrimidine itself, bis(diphenylphosphino)amine, pyrrolidinyldithiocarbamate,  tris(pyrazoyl-1-yl)borate. As a result, the authors have synthesized synthesized several mono- and binuclear Pd(II) complexes, some of which displaying unexpected structural arrangements.   

Although the reactivity of Pd(II) complexes with triphenylphosphine is not a subject of great novelty, this paper reports a complete and exhaustive work of ‘classic’ coordination/metallo-organic chemistry. The complexes obtained by the authors have been competently characterized by mass spectroscopy and 1H NMR. Most of the complexes have also been characterized by X-ray single crystal structure resolution (I am not, however, an expert of the technical aspects of X-ray crystallography).  Finally, the authors conclusions are well supported by the experimental data.

Therefore, I suggest publication of this paper in molecules.

I have just a minor revision to suggest. The names of the ligands used in the analysis of complex 1 reactivity should be also reported in the abstract, in order to facilitate the immediate comprehension of the results reported in this paper.     

Author Response

Q1:I have just a minor revision to suggest. The names of the ligands used in the analysis of complex 1 reactivity should be also reported in the abstract, in order to facilitate the immediate comprehension of the results reported in this paper.

A1: Line 15: The name of the ligand pyrimidinyl {pyrimidinyl = C4H3N2} has been added in the abstract.

We thank three reviewers’s comments for the improvement of the MS and We hope the revised MS is acceptable for publication in your journal.

Best Regards,

Dr. H.F. Wang

Reviewer 3 Report

The paper is interesting. However some minor points can be corrected. First in Abstract fix the IUPAC name of the ligands C4H3N2)= … -Hdppa =… and in the text at line 196 “dam” = ??

In the text correct dimmer as” dimer “ (twice)

boat-form geometry ??? in 3 and 4 … I cannot see any boat form. Better to indicate that   the mean planes through  Pd1-Pd2-n1-C1 and Pd1-Pd2-N3-C5 forms a dihedral angle of …xx °

It should be clearly indicated that complex 4 contains Pd(I) and this justify the short Pd-Pd distance

The degree of twisting  of the eight-membered ring containing the two palladium, four P…  how was calculated ? It not clear.

IN 3.2 it is not necessary the statement “The final residuals … R = 0.055 and Rw = 0.116 for 4, and R = 0.042 and Rw = 0.108 for 8.”, being these data in Table 1.

All hydrogens … thermal parameters 1.2  not 1-2

In Table 1, space group P1 should be P-1, although I cannot check since the authors did not provide CIF files, although in 3.2 is indicated that CIF files…are given in the supplementary material.

Why the S for complex 8 is so low ? (0.765) It should be close to 1.0, Was the weighting scheme correctly applied ?

In Conclusion and in synopsis “formative” should be” formation “

In synopsis correct as “The structure of complexes 3, 4, and 8 were determined by… “

Author Response

Q1: First in Abstract fix the IUPAC name of the ligands C4H3N2)= … -Hdppa =… and in the text at line 196 “dam” = ??

A1: Line 15, 18-19, 196: The IUPAC name of the pyrimidinyl {pyrimidinyl = C4H3N2}, Hdppa {Hdppa = N, N-bisdiphenyl phosphinoamine} and dam {dam = bis(diphenylarsino)methane} have been added in the abstract and text.

Q2:In the text correct dimmer as” dimer “ (twice)

A2:Line 130, 196: The two words ”dimmer” have been corrected to “dimer”.

Q3:boat-form geometry ??? in 3 and 4 … I cannot see any boat form. Better to indicate that  the mean planes through  Pd1-Pd2-n1-C1 and Pd1-Pd2-N3-C5 forms a dihedral angle of …xx °

A3:Line 133, 349: The boat-form geometry similar to those of complexes 3 and 8 have been published in 7 papers (Inorg. Chem. Commun. 2003, 6, 577-580., J. Chin. Chem. Soc. 2004, 51, 279-290, 493-498., J. Chin. Chem. Soc. 2007, 54, 553-558., J. Chin. Chem. Soc. 2008, 55, 109-114., Organometallics 2010, 29, 3397-3403., and J. Chin. Chem. Soc. 2009, 56, 718-724.). The sentence has been revised to “similar boat-form geometry”.

Q4:It should be clearly indicated that complex 4 contains Pd(I) and this justify the short Pd-Pd distance

A4:Line 206-208: The Pd-Pd separation of 2.6414 (6) Å of complex 4 is well within the range normally reported for Pd-Pd single bonds. It falls between those values observed for the unbridged complex Pd2(CNMe)62+ (2.5310 (9) Å) [17a] and bridged complexes Pd2Br2(dppm)2 (2.699 Å) [17b].

From the two references (Crystal and molecular structure of bis-µ-(bisdiphenylphosphinomethane) -dibromodipalladium(PdPd), a compound containing palladium(I). J. Chem. Soc., Chem. Commun. 1976, 12, 485-486. and Palladium-palladium .sigma.-bonds supported by bis(dimethylphosphino)methane (dmpm). Synthetic, structural, and Raman studies of Pd2X2(dmpm)2 (X = Cl, Br, OH). Inorg. Chem. 1985, 24, 3589-3593.), it is clear that the complex 4 contains Pd(I) and the sentence has been added in the revised MS (Line 202).

Q5:The degree of twisting  of the eight-membered ring containing the two palladium, four P…  how was calculated ? It not clear.

A5:Line211-212: The sentence has been rewritten to “The mean planes through Pd1-Pd2-P1-P2-N1 and Pd1-Pd2-P3-P4-N2 forms a dihedral angle of 32.662o.” for clearly.

Q6:IN 3.2 it is not necessary the statement “The final residuals … R = 0.055 and Rw = 0.116 for 4, and R = 0.042 and Rw = 0.108 for 8.”, being these data in Table 1.

A6:Line 340: The sentence “The final residuals of this refinement were R = 0.055 and Rw = 0.116 for 4, and R = 0.042 and Rw = 0.108 for 8.” has been deleted because these data are in Table 1.

Q7:All hydrogens … thermal parameters 1.2  not 1-2

A7:Line 338: All hydrogens were assigned isotropic thermal parameters “1-2” has been corrected to “1.2” Å2.

Q8: In Table 1, space group P1 should be P-1, although I cannot check since the authors did not provide CIF files, although in 3.2 is indicated that CIF files…are given in the supplementary material.

A8: P7, In Table 1, space group: P1 has been corrected to P-1.

Q9:Why the S for complex 8 is so low ? (0.765) It should be close to 1.0, Was the weighting scheme correctly applied ?

A9:P7, Table 1; P8, Table 2: The crystal data of complex 8 has been recheck and corrected (Independent rflns, no. of variables, Ra, Rwb, Sc, and Table 2 bond lengths and Bond angles) in the revised MS(in red color).

Q10:In Conclusion and in synopsis “formative” should be” formation “

A10:Line 352 and P15 in synopsis: The two “formative” words have been corrected to “formation”.

Q11:In synopsis correct as “The structure of complexes 3, 4, and 8 were determined by… “

A11: P15, In synopsis, the sentence has been corrected from “Complexes 3, 4, and 8 were determined by single-crystal X-ray diffraction analyses.” to “The structure of complexes 3, 4, and 8 were determined by single-crystal X-ray diffraction analyses.”

We thank three reviewers’s comments for the improvement of the MS and We hope the revised MS is acceptable for publication in your journal.

Best Regards,

Dr. H.F. Wang